# Empathy through the Pandemic: Changes of Different Emphatic Dimensions during the COVID-19 Outbreak

**DOI:** 10.3390/ijerph19042435

**Published:** 2022-02-20

**Authors:** Chiara Baiano, Gennaro Raimo, Isa Zappullo, Marialaura Marra, Roberta Cecere, Luigi Trojano, Massimiliano Conson

**Affiliations:** Department of Psychology, University of Campania Luigi Vanvitelli, 81100 Caserta, Italy; chiara.baiano@unicampania.it (C.B.); gennaro.raimo@unicampania.it (G.R.); isa.zappullo@unicampania.it (I.Z.); eyoml17@gmail.com (M.M.); roberta.cecere@unicampania.it (R.C.); luigi.trojano@unicampania.it (L.T.)

**Keywords:** empathy, social skills, theory of mind, social cognition, COVID-19

## Abstract

Growing evidence suggests that empathy is a relevant psychological trait to face the challenges imposed by the COVID-19 pandemic, but at the present very little is known on whether this multi-dimensional construct has been affected by the pandemic outbreak differently in its separate components. Here, we aimed at filling this gap by capitalizing on the opportunity of having collected data from different self-report measures and cognitive tasks assessing the main dimensions of empathy immediately before the beginning of the global pandemic and about one year later. The results showed a detrimental impact of the pandemic outbreak on empathic social skills but not on both cognitive (perspective-taking) and emotional empathy that instead significantly improved. Thus, reduced empathic social skills could be a weakness to be targeted in psychological interventions to help people cope with the mental health challenges related to COVID-19 pandemic, whereas the ability of understanding another’s mental states and emotions could represent a strength in dealing with the current long-lasting crisis.

## 1. Introduction

The COVID-19 outbreak, the economic losses associated with it, together with measures implemented to decrease the likelihood of contagion, such as social distancing, have resulted in severe psychological effects on the general population [1,2,3,4]. Furthermore, dissemination of information on social media about the virus has contributed to exacerbating emotional distress among the public [5]. Stress-related disorders such as anxiety, PTSD, and insomnia were found in the early phases of COVID-19 [6]; depressive symptoms were also observed in large sample studies [4,7] with increasing severity expected to be found as the social distancing measures were kept in the long term [8].

As the pandemic situation has lasted for two years worldwide, it is important to pay attention to people’s psychological well-being [9,10,11,12]. In this perspective, some psychological features could allow individuals to deal with the negative impact of the lockdown on psychological well-being and, also, to comply with public health rules [13]. Empathic abilities seem to be strictly related to mental health, as empathy deficits are present in several psychopathological conditions [14]. Moreover, recent data suggest that empathic abilities could play an important role both in supporting positive responses to stress during a pandemic [15] and in favoring compliance with pandemic-related measures [13,16,17]. However, specific empathic abilities, such as taking another person’s point of view and feeling of compassion and concern for others, could motivate individuals to adhere to policies as social distancing. On the other hand, personal distress—conceived as one’s own discomfort and anxiety when observing another’s negative experience—may not favor promotion of social distancing [16].

Empathy is indeed a multi-dimensional construct generally referring to the individual’s ability of understanding another’s mental states and emotions, and is thought to encompass distinct affective, cognitive, and social components [18,19]. Affective empathy refers to the degree to which a person indirectly experiences the feelings of another person or emotionally reacts to others’ psychological states [19]. Cognitive empathy, also described as ‘perspective taking’ or ‘theory of mind’ (ToM) [20], corresponds to the person’s ability to understand what another agent believes and feels by taking her/his point of view [21]. Social aspects of empathy (or ‘social skills’) imply active listening, turn-taking, sharing, and collaboration abilities [19].

Due to the centrality of empathy in dealing with the complex challenges caused by the COVID-19 pandemic and its role in mental well-being, it would be interesting to verify whether this psychological trait, albeit classically conceived as a stable one [19], may have been affected by the pandemic outbreak differently in its separate components. In this regard, a previous study [22] suggested that distinct aspects of empathy can differently change during the pandemic outbreak. Authors found that participants reported higher levels of cognitive empathy (perspective taking) but no change in affective empathy (empathic concern) a few months after the strict social confinement during the first wave of the COVID-19 pandemic in Spain [22].

Here, we aimed at evaluating the impact of the pandemic on the main dimensions of empathy within a sample of healthy individuals assessed before the COVID-19 outbreak and about one year after the implementation of policies to face the spreading of the coronavirus in Italy. The data before the outbreak were collected within a project aimed at evaluating the different components of empathy in young adults that started before the outbreak of the pandemic. We adopted several well-established self-report questionnaires and cognitive tasks to assess the main dimensions of empathy effectively. Indeed, after Brüne and Brüne-Cohrs [20], the use of multiple measures is recommended to overcome the current discrepancies in terminology and operational definitions (different constructs could share the same terminology and the same construct could have different acceptations, e.g., cognitive empathy has also been defined in terms of perspective taking or in terms of ToM). Moreover, our choice was also supported by the findings that the different components of empathy could be partially dissociable [19,21].

For the purposes of the present study, the same group of university students was assessed between October 2019 and first days of February 2020 (T0), and assessed again one year later, between December 2020 and February 2021 (T1). We could predict that the three main empathic domains we assessed here—i.e., affective empathy, cognitive empathy, and empathic social skills, changed from T0 to T1 differently. Following Baliyan et al. [22], we could expect that cognitive empathy would improve at T1 with respect to T0, while no changes were expected in affective empathy. No data were available to make predictions about changes in empathic social skills, but we could reasonably hypothesize that relevant limitations in sociability could negatively influence this dimension of empathy. 

## 2. Materials and Methods

### 2.1. Participants

A sample of 69 university students (mean age = 23.19 ± 2.82, range = 18–35; mean years of education =15.10 ± 1.73, range = 12–18; number of females = 49) was selected for participating in the research activities of the Laboratory of Developmental Neuropsychology, at the Department of Psychology, University of Campania ‘Luigi Vanvitelli’ (Caserta, Southern Italy). Participants were included in the study if they met the following inclusion criteria: (i) age between 18 and 35; (ii) lack of past diagnosis of neurodevelopmental disorders; and (iii) no clinical diagnosis of neurological, neuropsychiatric, or psychological disease. All participants were volunteers and spoke Italian as their native language.

The research was conducted after participants provided their written informed consent, which was approved by the Local Ethics Committee (code: N:30/2020) and performed in accordance with the ethical standards laid down in the 1964 Declaration of Helsinki.

### 2.2. Procedure

Within the lab’s research activities, the participants underwent an in-person assessment between October 2019 and first days of February 2020 (T0). About one year later, the participants were required to complete a second assessment (T1) between December 2020 and February 2021. The T1 assessment was conducted online by requiring participants to complete the same measures as at T0 through the Google Forms online platform (Google Inc., Mountain View, CA, USA). 

Hence, the T0 was conducted in a pre-pandemic period, while T1 covered a period in which, although in Italy there was not a strict lockdown, several activities pertaining the social life were still forbidden.

### 2.3. Measures

#### 2.3.1. Interpersonal Reactivity Index (IRI)

The IRI [23,24] is a 28-item (on a five-point Likert scale) self-report measure that detects empathic responsiveness through a multidimensional approach considering cognitive and affective empathy components. 

The IRI includes four factors: (i) the Fantasy scale, assessing the tendency to imaginatively transpose oneself into fictional situations (e.g., “I really get involved with the feelings of the characters in a novel”); (ii) the Perspective Taking scale, describing unplanned attempts to adopt others’ points of view (e.g., “I believe that there are two sides to every question and try to look at them both”); (iii) the Empathic Concern scale, indicating individuals’ feelings of compassion and concern for others (e.g., “I often have tender, concerned feelings for people less fortunate than me”); (iv) the Personal Distress scale, evaluating the extent to which an individual feels worry when exposed to others’ negative experiences (e.g., “Being in a tense emotional situation scares me”). Internal consistency of the Italian version of the IRI was acceptable (Cronbach’s alphas for each factor: FS = 0.74; PT = 0.64; EC = 0.63; PD = 0.75) [24].

#### 2.3.2. Empathy Quotient (EQ)

The EQ [25,26] was employed to measure empathy traits related to the ability to recognize others’ emotions and moods. The EQ provides a total score and three sub-scores (i.e., cognitive empathy, emotional reactivity, and social skills) [18,26]. Participants were administered the full 60-item EQ, with 40 questions tapping empathy (e.g., “I am quick to spot when someone in a group is feeling awkward or uncomfortable”) and 20 filler items (range score 0 to 80). The Italian version of the EQ [26] showed high reliability (Cronbach’s alpha = 0.79; test-retest at 1 month: Pearson’s r = 0.85).

#### 2.3.3. ‘Reading the Mind in the Eyes’ Test (‘Eyes Test’)

The ‘eyes test’ [27,28]. The ‘eyes test’ is considered an advanced ToM task, since participants are required to put themselves into the mind of the persons shown in the picture and attribute them a mental state. However, the task can also be considered an emotion recognition test [29]. 

The task consists of 36 black and white photographs showing the eyes area of adults, younger or older, male or female. Four possible responses related to emotional expressions are presented. Participants must choose the one of the four options that best describes the mental state or emotion represented in the picture. Moreover, participants are provided with a glossary which they can use in case of uncertainty about the meaning of the four options. The Italian version of the eyes test [28] showed a good reliability (Cronbach’s alpha = 0.77 obtained with Guttman split-half method).

#### 2.3.4. ‘Faux Pas’

The ‘faux pas’ test [30] was employed to evaluate a specific ToM ability, i.e., the ability to recognize involuntarily inappropriate social behavior. The task consists of 20 stories, 10 with a social error and 10 control items. After reading each story, participants must respond to questions aimed to verify: (i) whether the participants noticed the presence of the ‘faux pas’ (e.g., “Did someone say something that shouldn’t be said, or did someone say something inconvenient?”); (ii) if they considered the error not voluntary (e.g., “Did she/he remember it? Did she/he know it?”); (iii) if they understood the mental state of those who felt hurt or insulted (e.g., “How do you think she/he felt?”). Internal consistency of the ‘faux pas’ test has been estimated at 0.91 (Cronbach’s alpha) [31]. Moreover, the test–retest reliability reported in a previous study resulted adequate (Cronbach’s alpha = 0.83 over 3 months) [32].

### 2.4. Statistical Analysis

A MANOVA for repeated measures was conducted on IRI factors, EQ total scores and subscales, the eyes test, and the ‘faux pas’ total scores with time of the assessment (T0 vs. T1) as a within-subject factor. To control the increasing of type I error, each critical p-value was adjusted for multiple comparison by the Benjamini–Hochberg procedure (FDR) [33] with a false discovery rate of 0.05. The critical alpha level for all analyses was set <0.05. Analyses were performed using the Statistical Package for Social Sciences (SPSS Inc., version 25, Chicago, IL, USA).

## 3. Results

Results (Table 1) showed a significant main effect of time of the assessment on IRI Perspective Taking, F(1,68) = 82.25, *p* < 0.001, *η*^2^*_p_* = 0.547, IRI Fantasy, F(1,68) = 88.17, *p* < 0.001, *η*^2^*_p_* = 0.565, and Empathic Concern, F(1,68) = 161.8, *p* < 0.001, *η*^2^*_p_* = 0.704, always with higher scores at T1 with respect to T0. These results mean that, one year after the pandemic outbreak, participants showed a greater tendency to adopt other’s point of view (Perspective Taking), to transpose oneself into fictional situations (Fantasy), and to be concerned and compassionated about other people (Empathic Concern). No significant effect of time of assessment was found on Personal Distress, F(1,68) = 0.414, *p* = 0.552, *η*^2^*_p_* = 0.006.

Moreover, significant main effects of time were found on EQ total score, F(1,68) = 9.81, *p* = 0.003, *η*^2^*_p_* = 0.126, EQ emotional reactivity subscale, F(1,68) = 313.87, *p* < 0.001, *η*^2^*_p_* = 0.822, and EQ social skills, F(1,68) = 225.47, *p* < 0.001, *η*^2^*_p_* = 0.768, with higher scores on EQ total score and emotional reactivity, and lower scores on EQ social skills at T1 with respect to T0. These findings mean that, one year after the pandemic outbreak, participants reported higher levels of empathy in general (EQ total score) and emotional empathy (EQ emotional reactivity), while they reported a lower tendency to perceive and understand other’s mental states through social cues (EQ social skills). No effect was found on EQ Cognitive Empathy, F(1,68) = 2.02, *p* = 0.159, *η*^2^*_p_* = 0.029. 

Finally, results showed a significant effect of time of assessment on the ‘eyes test’, F(1,68) = 4.49, *p* = 0.038, *η*^2^*_p_* = 0.062, with lower scores at T1 with respect to T0. This finding means that, one year after the pandemic outbreak, participants showed a lower ability to infer other’s mental states from other persons’ eyes. Instead, no effect was found on ‘faux pas’, F(1,68) = 1.42, *p* = 0.237, *η*^2^*_p_* = 0.021.

The FDR correction confirmed the significance of all the *p*-values with the only exception being the ‘eyes test’ score (FDR adjusted *p*-value = 0.054).

## 4. Discussion

The present study showed that, one year after the pandemic outbreak, healthy young individuals had lower empathic social skills, whereas the cognitive and emotional aspects of empathy, as well as tendency to identify with fictional characters, increased.

Stronger perspective-taking abilities are associated to a higher ability to evaluate situations rationally and obtain healthier points of view [34]. Moreover, both perspective taking and empathic concern have been related to multiple aspects of trait mindfulness [35], which is a protective factor from worry and anxiety during the COVID-19 outbreak [36]. On the other hand, a difficulty in completing tasks implying perspective taking has been observed in individuals scoring high on a measure of depressive symptoms [37]. 

On this basis, one could speculate that direct or indirect (through social media) experience of other people dealing with the complex requirements imposed by the pandemic could favor the tendency to evaluate situations from others’ points of view. This might also account increased score of IRI Fantasy, while leaving unchanged IRI personal distress that measures a self-oriented state of personal distress in response to another’s negative state, a response that has been considered not emphatic in itself [25].

It is important to underscore that IRI perspective taking changed through the two assessment periods, whereas cognitive empathy, as measured by EQ cognitive empathy subscale, did not. Although EQ cognitive empathy implies perspective-taking, it represents a more general construct involving understanding of both others’ mental states and emotions. Indeed, in Lawrence et al.’s [19] study investigating reliability and validity of the EQ, the lack of association between the cognitive empathy factor and IRI perspective-taking suggested that different types of mental state attribution are assessed by the two measures. Significant increases of emotional empathy at T1 on the IRI Empathic Concerns and EQ emotional reactivity, were not expected. Indeed, Baliyan et al. [22] assessed a group of young adults before and a few months after the strict social confinement during the first wave of the COVID-19 pandemic in Spain. Their results showed increased cognitive empathy (perspective taking) which contributed to perceived self-efficacy but no change in affective empathy (empathic concern). Notwithstanding this discrepancy, increased emotional empathy is in line with data showing that greater emotional empathy can predict acceptance of lockdown measures [13], also being positively associated with self-reported social distancing behavior [16]. 

Relevantly, here we found at T1 lower empathic social skills and a trend towards significantly lower scores on the ‘eyes test’, a formalized cognitive task providing a measure of understanding of social situations correlating with EQ social skills [19]. Such findings might suggest that social distancing, isolation, the use of face masks, and possibly extensive use of home-based communication technologies could have negatively impacted on social skills. These complex changes in everyday life imposed by the pandemic likely decreased the ability to perceive and understand other’s mental states through social cues, while leaving unmodified the capacity to make cognitive inferences about social situations, as shown by the lack of significant changes on the ‘faux pas’ test. This interpretation would be consistent with data showing that recognition of emotional facial expressions is lowered when judging faces wearing masks [38], although this result has not been reported consistently [39]. Thus, it is possible to speculate that the individual’s capacity to pay attention to all the social cues implied in social interactions suffered most (and likely is still suffering now) from measures imposing social distancing.

The present findings have potential clinical implications. Given that the pandemic situation has lasted for nearly two years as of writing, the need to pay more attention to public mental health is growing [9,10,11,12]. Mental health is related to empathic abilities, as deficits of empathy can represent a neuropsychological mechanism involved in different emotional disorders [14,40], and dysfunctions of cognitive empathy seem to contribute to impaired social functioning in psychopathology [20]. In this view, the present results would highlight the need to consider the different impact of COVID-19 pandemic on the main dimensions of empathy in planning interventions of psychological support.

The present study has some limitations. First, we assessed university students who can show distinct patterns personality traits including empathy [41,42]. Additionally, due to the small size of the sample and the unbalanced number of women and men we could not test whether sex could moderate the effect of time of measurement on empathic abilities that are known to differ between sexes [19,25]. Moreover, T0 was conducted in-person while T1 was conducted online, thus the distinct settings could have differently affected participants’ responses at the two times of assessment. Indeed, several studies highlighted differences among participants in the accessibility of the medium and in the technical skills, as well as the different effects of the online and the face-to-face settings on engaging the participant in the relationship with the researcher [43,44]. These methodological concerns need to be considered as factors impacting on the generalizability of the present findings. Finally, it could be interesting to directly assess the impact of the increased use of home-based communication technologies on empathic abilities, a point that was not tackled in the present study (for instance see Watson et al. [45]).

## 5. Conclusions

Here, we investigated whether the COVID-19 pandemic affected the separate components of empathy to a different extent. We observed that, about one year after the global pandemic outbreak, empathic social skills were impacted negatively whereas perspective-taking and emotional empathy increased. The capacity to empathize with and share another’s emotion could represent a person’s strength, while reduced empathic social skills could be a weakness to be targeted in psychological interventions to help people cope with the mental health challenges related to COVID-19 pandemic [3,46].

## Figures and Tables

**Table 1 ijerph-19-02435-t001:** Effect of time on empathy measures.

	T0	T1	F(1,68)	*p*	*η* ^2^ * _p_ *
Interpersonal Reactivity Index (IRI)					
Perspective Taking	23.25 ± 3.00	26.93 ± 4.25	82.25	<0.001 *	0.547
Fantasy	21.33 ± 3.15	26.06 ± 4.28	88.17	<0.001 *	0.565
Empathic Concern	21.30 ± 2.39	28.28 ± 4.35	161.8	<0.001 *	0.704
Personal Distress	19.49 ± 3.27	19.83 ± 4.31	0.414	0.522	0.006
Empathy Quotient (EQ)					
Total score	44.48 ± 10.71	47.43 ± 11.35	9.81	0.003 *	0.126
Cognitive empathy	13.67 ± 4.21	14.35 ± 4.31	2.02	0.159	0.029
Emotional reactivity	5.52 ± 2.24	14.62 ± 4.27	313.87	<0.001 *	0.822
Social skills	13.10 ± 4.45	5.74 ± 2.73	225.47	<0.001 *	0.768
‘Eye test’	26.48 ± 2.97	25.61 ± 3.12	4.49	0.038	0.062
‘Faux pas’	41.01 ± 9.11	39.51 ± 11.34	1.42	0.237	0.021

Note: Asterisk means significant *p*-values after FDR correction (*p* < 0.004).

## Data Availability

Data are available upon request to the corresponding author.

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
