# Peer review of "Empathy through the Pandemic: Changes of Different Emphatic Dimensions during the COVID-19 Outbreak"

_ijerph, 2022, doi:10.3390/ijerph19042435_

Round 1

Reviewer 1 Report

The Authors of the manuscript raised a very important issue of empathic behavior in the pandemic era. The manuscript is clearly written, however there are a few points that need to be elaborated on.

  1. Introducion - in this section you will find the items that correspond and should be included in the Discussion (lines 57-63). In this part of the manuscript, I miss more information on the types of empathy studied, what are the differences between them, how they can be developed, etc.
  2. Lines 79-80 - does the information provided refer only to women or to the entire study population? Because such a presentation of them is not entirely readable.
  3. Line 84 - on what basis was it determined that participants were showing "typical cognitive development"
  4. Lines 114-115 - what types of questions were in the test. One of such questionnaires should be included in additional materials.
  5. Table 1 shows the results, but it is very difficult to interpret them in the Results issue. These data (from the text) should also be tabulated so that it is possible to identify exactly which differences are statistically significant and which are not. Unfortunately, this part of the research is presented in a rather chaotic way and should be improved.

Author Response

Reviewer #1

General Comment. The Authors of the manuscript raised a very important issue of empathic behavior in the pandemic era. The manuscript is clearly written, however there are a few points that need to be elaborated on.

Response. We thank the Reviewer for this positive comment on our manuscript.

Comment 1. Introduction - in this section you will find the items that correspond and should be included in the Discussion (lines 57-63). In this part of the manuscript, I miss more information on the types of empathy studied, what are the differences between them, how they can be developed, etc.

Response. As the Reviewer suggested, we moved part of the lines 57-63 to the Discussion. We also added information about the construct of empathy in the Introduction section, underscoring the relevance of this psychological trait with respect to mental health, especially in the context of the pandemic outbreak.

Comment 2. Lines 79-80 - does the information provided refer only to women or to the entire study population? Because such a presentation of them is not entirely readable.

Response. In the revised version we better specified that information refers to the entire sample in the following way: “A sample of 69 university students (mean age = 23.19 ± 2.82, range = 18-35; mean years of education =15.10 ± 1.73, range = 12-18; number of females = 49) was selected for participating in the research activities of the Laboratory of Developmental Neuropsychology, at the Department of Psychology, University of Campania ‘Luigi Vanvitelli’ (Caserta, Southern Italy).”

Comment 3. Line 84 - on what basis was it determined that participants were showing "typical cognitive development"

Response. In the revised version of the manuscript, we specified that, to be included in the study, participants were not expected to have been diagnosed with neurodevelopmental disorders in their past history.

Comment 4. Lines 114-115 - what types of questions were in the test. One of such questionnaires should be included in additional materials.

Response. Due to copyright reasons, it was not possible for us to add questionnaires in supplementary materials. However, following the Reviewer’s comment, in the revised version of the manuscript, we have added some items to provide examples of the types of questions included in EQ and IRI.

Comment 5. Table 1 shows the results, but it is very difficult to interpret them in the Results issue. These data (from the text) should also be tabulated so that it is possible to identify exactly which differences are statistically significant and which are not. Unfortunately, this part of the research is presented in a rather chaotic way and should be improved.

Response. We thank the Reviewer for the suggestion. In the revised version of the manuscript, we tabulated all data to facilitate readability.

Reviewer 2 Report

This is an interesting study that is well-constructed and -organized.

It just needs some areas for elaboration.

The methodology portion where the tests are described could be broken down a little further. Why are all these tests needed to test for empathy? 

Obviously the authors began the study without knowing that a pandemic would occur; was the idea just to see if time reduces empathy or might something else have been the goal of the research?

The results badly need to be broken down more clearly and in a more coherent fashion. There is just a short paragraph with numbers thrown at the reader. Each result should be explained and, in turn, it's implications should be discussed. I think the results are run through too quickly making the conclusion seem less certain. Some of the numbers go up over time, others go down. Are we to make of this that empathy changed in all dimensions? The faux-pas test certainly didn't. How can we support the authors' conclusions without a more careful and deliberate presentation and explanation of the results?

Author Response

Reviewer #2

General Comment. This is an interesting study that is well-constructed and -organized. It just needs some areas for elaboration.

Response. We thank the Reviewer for this positive comment on our manuscript. We elaborated the revised version as the Reviewer suggested.

Comment 1. The methodology portion where the tests are described could be broken down a little further. Why are all these tests needed to test for empathy? 

Response. As suggested by the Reviewer, we separated the paragraph in specific subsections, each dedicated to a specific measure used for the study. Moreover, in the present study we adopted several well-established self-report empathy measures in order to cover all dimensions of empathy, which resulted to be partially dissociable (Healey & Grossman, 2018; Lawrance et al., 2004). Moreover, it should be also underlined here that, in empathy literature, terminology and operational definitions can differ when referring to the same construct, or different constructs can also share the same terminology. For example, cognitive empathy has also been defined in terms of perspective taking or in terms of ToM. For this reason, and within these terminological contexts, the use of different self-report measures and cognitive tests was recommended (Brüne and Brüne-Cohrs, 2006). We specified this point in the revised Introduction section.

Comment 2. Obviously the authors began the study without knowing that a pandemic would occur; was the idea just to see if time reduces empathy or might something else have been the goal of the research?

Response. Data available at T0 were collected within a project aimed at evaluating the different components of empathy in a sample of university students. After the pandemic outbreak, an increasing number of studies suggested that empathy was strongly involved in individuals’ psychological responses to covid-19 pandemic, and interestingly some of these studies provided some hints on the possibility that not all the dimensions of empathy could be related in the same way to individuals’ psychological functioning during the pandemic. Thus, we realized that it could be important to understand whether the pandemic impacted on this important psychological feature, and even more importantly, due to its multifaceted nature, whether the different facets of empathy met the same fate. Hence, the main aim of the present study was conceived as reported.

Comment 3. The results badly need to be broken down more clearly and in a more coherent fashion. There is just a short paragraph with numbers thrown at the reader. Each result should be explained and, in turn, it's implications should be discussed. I think the results are run through too quickly making the conclusion seem less certain. Some of the numbers go up over time, others go down. Are we to make of this that empathy changed in all dimensions? The faux-pas test certainly didn't. How can we support the authors' conclusions without a more careful and deliberate presentation and explanation of the results?

Response. We thank the Reviewer for this suggestion. In the revised version of our manuscript, we reframed the Results section describing and explaining the results of the study in detail. Moreover, we better specified all data in Table 1 to facilitate results’ understanding and readability. 

Reviewer 3 Report

This was an interesting and well-designed study, and while it seems like an obvious area for inquiry, it is the first I have seen of anything like it.

In the abstract, "in the shoes," is an idiommatic phrase (cliche) and I would recommend revising. Also lines 27-28: "dissemination of information on death from social media and a large amount of misinformation on the virus" is clunky and potentially confusing, like social media was the bringer of death, which is somewhat true, but not what I think was intended, which was ON social media versus from it.

The introduction could flow more smoothly, better integrating and  synthesizing  the "ground" of the pandemic and the "figure" of empathy.

Overall some of the sentences were run-on with inappropriate use of commas (splicing) which could be remedied by using shorter sentences and better pacing of information.

I was a little surprised that while the potential effects of masks wre discussed, no mention or accounting was made of video use during the pandemic. There is research to suggest that, for example, Zoom images impose greater intimacy, to the point of potentially prompting aggressive reactions.

Author Response

Reviewer #3

General Comment. This was an interesting and well-designed study, and while it seems like an obvious area for inquiry, it is the first I have seen of anything like it.

Response. We thank the Reviewer for this positive comment on our manuscript.

Comment 1. In the abstract, "in the shoes," is an idiomatic phrase (cliche) and I would recommend revising. Also lines 27-28: "dissemination of information on death from social media and a large amount of misinformation on the virus" is clunky and potentially confusing, like social media was the bringer of death, which is somewhat true, but not what I think was intended, which was ON social media versus from it.

Response. In the revised version of the abstract, we rephrased the sentence as follows: “The reduced empathic social skills could be a weakness to be targeted in psychological interventions to help people cope with the mental health challenges related to COVID-19 pandemic, whereas the ability to understand and share another’s emotion could represent a strength in dealing with the current long-lasting pandemic”. We also rephrased lines 27-28: “Furthermore, dissemination of information on social media about the virus contributed to exacerbate emotional distress among the public”.

Comment 2. The introduction could flow more smoothly, better integrating and synthesizing the "ground" of the pandemic and the "figure" of empathy.

Response. As suggested by the Reviewer (see also Reviewer #1, Comment #1), in the revised Introduction section we provided more information about the construct of empathy, also underscoring the relevance of this psychological trait with respect to mental health, especially in the context of the pandemic outbreak.

Comment 3. Overall, some of the sentences were run-on with inappropriate use of commas (splicing) which could be remedied by using shorter sentences and better pacing of information.

Response. Following Reviewer’s comment, we revised phrasing throughout the manuscript.

Comment 4. I was a little surprised that while the potential effects of masks were discussed, no mention or accounting was made of video use during the pandemic. There is research to suggest that, for example, Zoom images impose greater intimacy, to the point of potentially prompting aggressive reactions.

Response. We thank the Reviewer for the interesting point. In the revised Discussion section, we reported that it could be interesting to directly assess the impact of the increased use of home-based communication technologies on empathic abilities, a point that had not been tackled with in the present study (for instance see [Watson et al., 2021]).